# Maturity and Strength Development of Mortar with Antifreezing Admixture at Temperatures Lower than 0 °C

**DOI:** 10.3390/ma12193172

**Published:** 2019-09-27

**Authors:** Hyeonggil Choi, Yukio Hama, Madoka Taniguchi

**Affiliations:** 1School of Architecture, Kyungpook National University, Daegu 41566, Korea; hgchoi@knu.ac.kr; 2College of Environmental Technology, Graduate School of Engineering, Muroran Institute of Technology, Muroran, Hokkaido 050-8585, Japan; 3Environmental Engineering Division, Northern Regional Building Research Institute, Hokkaido Research Organization, Asahikawa, Hokkaido 078-8801, Japan; madoka@hro.or.jp

**Keywords:** blast furnace slag cement, antifreezing admixture, maturity, equivalent age, strength, cold-weather concreting

## Abstract

This study investigated the effect of temperature and time at temperatures lower than 0 °C on mortar mixed with antifreezing admixture to determine the temperature–time function with the aim of expressing the effect universally. As a result, the maturity equation for temperatures lower than 0 °C proposed in previous studies was verified to be applicable to type-B blast furnace slag cement. The applicability of this equation at temperatures lower than 0 °C had not been investigated hitherto. The strength development attributable to the effect of the antifreezing admixture can be expressed as the reference temperature, and the reduction in the chemical potential of water chemical potential reduction was found to depend on the reaction rate. A new maturity equation for temperatures lower than 0 °C was proposed considering the effect of the antifreezing admixture.

## 1. Introduction

The use of blast furnace slag cement admixed with ground granulated blast-furnace slag has been expanded to reduce environmental loads and employ industrial byproducts efficiently [1,2,3]. Although blast furnace slag cements have many advantages such as low heat, increased long-term strength, flame interruption performance, and suppression of alkali aggregate reaction, they have drawbacks such as delayed early strength development and strong temperature dependence. These drawbacks necessitate considerable care, such as measures to prevent early frost damage during cold-weather concreting [1,2,3,4].

Antifreezing admixtures are used to prevent early frost damage in concrete during cold-weather concreting [5,6,7]. Concrete admixed with antifreezing admixture exhibits strength-enhancement effects at low temperatures because the accelerator hastens hardening and lowers the freezing point. Moreover, such a concrete mixture can overcome the drawback of early strength reduction that is characteristic of blast furnace slag cements during cold-weather construction. Thus, many studies have analyzed concrete samples prepared by combining blast furnace slag cement and antifreezing admixture from the viewpoints of reducing environmental load and cold-weather concreting [4,8].

Generally, a maturity is used to estimate the strength of concrete in cold-weather construction [5,9,10,11]. A maturity equation for estimating the strength of concrete at temperatures lower than 0 °C has been proposed in the literature [5,11]. However, this empirical equation is limited to ordinary Portland cement (OPC) and fly ash cement Type B (FB). It cannot be applied to concrete samples containing blast furnace slag cement Type B (BB) or antifreezing admixture.

Thus, in the present study, we conduct a mortar experiment to investigate the applicability of the aforementioned maturity equation to mortar containing BB or BB admixed with antifreezing admixture at temperatures lower than 0 °C. In addition, we propose a maturity equation for mortar samples containing the antifreezing admixture.

## 2. Experimental Design and Methods

### 2.1. Experimental Plan and Materials

Table 1, Table 2, Table 3 and Table 4 present the experimental plan, materials used, materials properties, and the Sieve analysis results of sand herein. Two types of cement, namely, OPC and BB, were employed to prepare the experimental specimens. As the antifreezing admixture, the main component employed was an inorganic nitrogenous compound of nitrite or nitrate, which consists of only a frost-resistant component without any water-reducing or air-entrained component. In the experiment, the mortar was mixed according to the Test Methods for the Determination of Strength in Cement (ISO 679) with a water-to-cement weight ratio of 0.5 and a cement-to-fine aggregate weight ratio of 1:3. To these samples, 0%, 2%, 4% and 6% of antifreezing admixture per 100 kg of cement were added.

### 2.2. Experimental Method

After the mortar was poured, cylindrical specimens measuring Ø 50 mm × 100 mm were fabricated and demolded at an age of 1 day. The compressive strengths of two specimens at a certain age were measured in accordance with JIS A 1108: the strength of a specimen that underwent water curing at 20 °C and sealed curing at 20 °C was considered the reference strength, and the strength of a specimen that was subjected to sealed curing at a temperature lower than 0 °C (−2, −5, and −10 °C) was considered as a reference value for cold-weather concreting [12]. A thermocouple was embedded inside each specimen at its center, and the measured internal temperature of the specimen was recorded on an hourly basis with a data logger.

## 3. Experimental Results and Discussion

### 3.1. Relationship Between Age and Compressive Strength

Figure 1 and Figure 2 show the relationships between age and compressive strength for OPC and BB. The figures show that strength development in both OPC and BB deteriorates under low-temperature conditions, and that the strength development effect improves with the addition of increasing quantities of the antifreezing admixture. There was no significant difference in strength development at the temperatures of −10 and −5 °C. This was due to the uncontrolled period of temperature gradually in the constant temperature chamber set to −10 °C. However, because the maturity was investigated considering the actual temperature history inside the mortar, this lack of difference in strength development did not cause a significant problem when deriving the maturity equation. In addition, nearly identical levels of strength development were observed for the mortar samples subjected to sealed curing and water curing at 20 °C. This indicated that the decrease in internal humidity owing to hydration during sealed curing did not influence the strength development of the specimens.

### 3.2. Maturity and Compressive Strength

It is generally acceptable to compute the maturity with Equation (1) for predicting the strength development of concrete. The maturity equation applicable at temperatures lower than 0 °C, as proposed in previous studies, is given in Equation (2) [5,11].
Case of T ≧ 0 *M* = ∑(*T* + 10)*Δt*(1)
Case of T < 0 *M* = ∑10 × exp(−0.60 × (−*T*)^0.74^)*Δt*(2)
where *M* denotes the maturity (°D·D), T the temperature (°C), and *Δt* the time interval (days).

Figure 3 and Figure 4 show the relationship between the compressive strengths of OPC and BB and their maturity at room temperature and at temperatures lower than 0 °C, as computed using Equations (1) and (2), respectively. The strength development of the OPC and BB specimens without the addition of the antifreezing admixture (OPC-0, BB-0) was well verified using the above maturity. The strength development of the mortar specimen composed of BB could be well predicted with Equation (2). This indicates that the existing maturity equation can be applied.

The compressive strength of the specimens containing the antifreezing admixture was higher than that of the reference curing specimen, and the strength deviation increased as the quantity of the antifreezing admixture added increased.

The relationship between the measured compressive strength and the compressive strength calculated using the maturity obtained with Equations (1) and (2) was investigated for the OPC and BB specimens containing the antifreezing admixture to verify the strength deviation caused by the addition of the antifreezing admixture. The results of this investigation are shown in Figure 5 and Figure 6. The correlation coefficients of OPC and BB are 0.93 and 0.92, respectively. Because the added antifreezing admixture lowered the freezing temperature, the measured strength exceeded the calculated strength. Thus, the effect of the antifreezing admixture should be considered in the maturity equation. Specifically, the strength of the mortar containing the antifreezing admixture can be estimated efficiently at temperatures lower than 0 °C by considering the freezing-point-lowering effect of the antifreezing admixture in the maturity equation.

### 3.3. Temperature Dependence of Strength Development

The strength development of concrete can be expressed in terms of the temperature–time function. That is, the effect of temperature and time can be converted into an age under the standard temperature condition. Several methods have been proposed for plotting the strength development curve. Two of the most widely used methods are maturity and equivalent age [5,9]. Among them, a previous study conducted an investigation based on measured data of concrete samples admixed with cement’s chemical additives that were recently used to propose a standard curve of strength development that could respond to a wide range of water–cement ratios according to the high-strength concretes [10]. Therefore, studies have investigated the suitability of the logistic and the Gompertz curves, which exhibit the strength development process of concrete in terms of the maturity as a temperature–time function. The results indicate that the Gompertz curve can be used to estimate the strength development of concrete more efficiently than the logistic curve [10,13]. Thus, in the present study, the Gompertz curve is used to estimate the strength development of concrete [5,14].
*F* = *F_inf_* × exp(*a* × *T ^b^*)
(3)
where *F* denotes the compressive strength for the temperature–time function *T* (N/mm²); *F_inf_* denotes the final compressive strength achieved (N/mm²); a and b denote test integers calculated with the nonlinear least squares method because *T* was divided into the maturity and the equivalent age to ensure compatibility with the actual test results. The results are presented in Table 5. The subsequent investigations herein employ these calculated coefficients.

The strength development of concrete is highly contingent on the hydration reaction of cement and water. The temperature dependence of the hydration reaction rate is expressed in terms of the equivalent age with the Arrhenius equation [15]. The chemical potential of water deteriorates significantly under drying and freezing conditions. It is necessary to consider this deterioration in the chemical potential of water when it reacts with cement at temperatures lower than 0 °C, and these temperatures are investigated in the present study. In a past study, the hydration of cement was modeled based on its apparent hydration rate, which was measured by considering the bound water ratio of cement paste, and the rate of the hydration reaction was discussed [16]. The results showed that the reaction rate of hardened cement paste in a drying or freezing environment decreased owing to a decrease in the chemical potential of the water reacting with the cement. This decrease was expressed as follows by incorporating the decrease in the chemical potential of water into the Arrhenius equation [15,16].
*k_T_* = *A*exp(−*E* + *αμ*/*RT*)
(4)
where *k_T_* denotes the reaction rate integer at temperature *T* (K); A and α denote experiment integer; *E* denotes apparent activation energy (J/mol); *μ* denotes the chemical potential of water (J/mol); *R* denotes the gas constant (J/mol·K).

The apparent activation energy *E*, which represents the temperature dependence, is needed to calculate the reaction rate integer *k_T_* in Equation (4). We calculated E using the following equation that considers the replacement ratio of ground granulated blast furnace slag [10].
*E* = 32.2 + 0.4*r*(5)
where *r* denotes the replacement ratio (%) of ground granulated blast furnace slag.

The equivalent age, which is a temperature–time function obtained using Equation (5), can be expressed as follows in terms of the reaction rate integer *k_T_* and the reference temperature *K_rf_* [11].
*t_e_* = ∑(*k_T_/k_rf_*)·*Δt* = −(exp((∑*E* + *αμ*/*RT*) − (−*E*/*RT_rf_*)))·*Δt*(6)
where *t_e_* denotes the equivalent age (days), *K_rf_* the reaction rate integer at the reference temperature *T_rf_(K)*, and *Δt* the time interval.

Figure 7 shows the relationship between the measured strength and the calculated strength. To determine the suitability of the apparent activation energy value that represents temperature dependence, a relationship with the strength calculated using Equation (3) was derived using the measured strengths of OPC and BB mortars and the equivalent age computed using Equation (6). Because the test to calculate the strength of the mortar specimens was conducted using the sealed-cured specimens, the effect of a reduction in the chemical potential of water owing to changes in internal humidity caused by the hydration reaction should be considered. However, as mentioned above, no significant difference was found between the strength-development trends of the specimens subjected to water curing and sealed curing at 20 °C, indicating that the reduction effect was absent. The coefficients of determination of the measured and calculated strengths of the individual specimens prepared with OPC and BB were 0.99, indicating a good correlation. Thus, the values obtained using Equation (5) have been used as the E values of the respective types of cement in the subsequent investigations.

### 3.4. Effects of Freezing and Antifreezing Admixture

#### 3.4.1. Effect of Reduction in Chemical Potential of Water

In the concrete admixtures without the antifreezing admixture, the chemical potential of water decreased, which affected the strength development of the admixtures. The change in the chemical potential of water *μ* (J/mol) due to freezing, as in Equation (4), can be calculated using the following equation [17]:*μ* = −58.73 × (−*T*’)^0.58^(7)
where *T*’ denotes temperatures lower than 0 °C. If *T*’ ≧ 0, then it satisfies *μ* = 0.

The optimum value of the experiment integer *α* in Equation (4) was calculated using the nonlinear least squares method and the strength development curve, made by the Gompertz curve calculated through 20 °C curing using the curing results at the sub-zero temperature condition of OPC and BB without adding the antifreezing admixture. The results are presented in Table 6.

Figure 8 shows the correlations between the equivalent age and the measured strength considering freezing, as calculated using Equation (6) and the experiment integer α, for OPC and BB. The respective correlation coefficients are 0.98 and 0.99 These results indicate that the equivalent age can well express the effect of freezing. Based on this result, the effect of the antifreezing admixture is investigated.

#### 3.4.2. Coefficient of the Effect of Antifreezing Admixture

Antifreezing admixture causes accelerated hardening and lowers the freezing temperature. Here, considering the reference temperature *T_rf_* and reduction in the chemical potential of water *α*, the effect, *α*, of the antifreezing admixture is expressed with the following equation for the admixture containing the antifreezing admixture.
*T_rf_* = 273.15 + *cx*(8)
*α*’ = *α* + *dx*(9)
where *α*’ denotes the experiment integer when the antifreezing admixture is added, *α* denotes the experiment integer when the antifreezing admixture is not added, *c* and *d* are experiment integers, and *x* denotes the quantity (ℓ) of the antifreezing admixture added per 100 kg of cement. Here, a common value of c was calculated for OPC and BB to consider the effect of a decrease in the freezing temperature and the effect of the apparent activation energy based on the type of cement. The optimal value of c was calculated using the nonlinear least squares method for each value of d for OPC and BB. The results are presented in Table 7. The equivalent age equation, including the effect of the antifreezing admixture, can be obtained by calculating the experiment integers.

### 3.5. Maturity Equation for Temperatures Lower than 0 °C Considering the Effect of Antifreezing Admixture

In this section, the maturity equation that can express the strength development of mortar at temperatures lower than 0 °C by considering the effect of the antifreezing admixture is derived using the experimental results obtained with mortar samples containing the antifreezing admixture. The equivalent age equation described in Section 3.3 and Section 3.4 can serve as a temperature–time function that expresses the strength development efficiently at temperatures lower than 0 °C considering the effect of the antifreezing admixture.

Here, based on Equation (2), which gives the temperature–time function in the absence of the antifreezing admixture [5,11], a temperature–time function that can be expressed with variables such as temperature and addition of the antifreezing admixture is regarded as a logarithmic value of the equivalent age, as given below:*ln*(*t_e_*) = (−0.60 + *βx*) × (−*T_c_* − *ex*)^(0.74 + *γx*)^(10)
where *T_c_* denotes temperature (°C); *β*, *γ*, and e are experiment integers; and *x* denotes the quantity of the antifreezing admixture (ℓ) added per 100 kg of cement.

The optimal values of the experiment integers *β*, *γ*, and *e* were calculated by applying Equation (10) to the test results of the mortar specimens containing the antifreezing admixture. This equation can be used to obtain the equivalent age as a temperature function at temperatures lower than 0 °C, in which the lowering of the freezing temperature by the antifreezing admixture is reflected against the reference temperature.

However, generally, maturity is used as the temperature–time function above 0 °C during cold-weather concreting. Thus, the use of maturity rather than the equivalent age, even at temperatures lower than 0 °C, when the antifreezing admixture is added can be convenient for predicting the strength of concrete during cold-weather concreting. The existing maturity equations can possibly be applied together by modifying Equation (10) to convert the age of one-day curing at 0 °C into the maturity of 10 °DD when the antifreezing admixture is not added.

Based on this rationale, the maturity computed considering the lowering of the freezing temperature upon the addition of the antifreezing admixture can be expressed with the following equation, thereby investigating the comparison with the experiment results.
*M_f_* = ∑(10 − *ex*) × exp{(−0.60 + *βx*) × (−*T_c_* − *ex*)^(0.74 + *γx*)^}·*Δt*(11)
where *M_f_* denotes the maturity (°C) at temperature *T_c_*, and *x* denotes the quantity (ℓ) of the antifreezing admixture added per 100 kg of cement. The optimum values of the coefficients *β*, *γ*, and *e* were determined with the nonlinear least squares method to make the relationship with the maturity matched with the experimental results with regard to each addition of the antifreezing admixture calculated by Equation (11) and compressive strength. The computed optimal values are *β* = 0.038, *γ* = −0.038, and *e* = 0.28.

Using the above results, the maturity equation considering the addition of the antifreezing admixture can be obtained as follows:Case of *T* ≧ −0.28*x**M* = ∑(*T* + 10)*Δt*(12)
Case of *T* < −0.28*x**M_f_* = ∑(10 − 0.28*x*) × exp{(−0.60 + 0.038*x*) × (−*T_c_* − 0.28*x*)^(0.74 − 0.038*x*)^}·*Δt*(13)
where *M_f_* denotes the maturity (°C) at temperature *T_c_*, *Δt* denotes the time interval (days), and *x* denotes the quantity (ℓ) of the antifreezing admixture added per 100 kg of cement.

The relationships between maturity and compressive strength for OPC and BB, calculated using Equations (12) and (13) in this study, are shown in Figure 9 and Figure 10, respectively. OPC-0 and BB-0, in which the antifreezing admixture was not added, satisfied *x* = 0. Thus, it is the same equation as the existing Equation (2). When the antifreezing admixture was added, the strength development of mortar could be expressed accurately by using the maturity given in Equation (13), as verified by the figures.

Figure 11 and Figure 12 show the relationship between the measured strength of each cement and the strength calculated considering the maturity at temperatures lower than 0 °C by considering the effect of the antifreezing admixture. This relationship verifies Equations (12) and (13). The coefficients of determination were 0.98 for OPC and 0.97 for BB, which are higher than the results in Figure 6 and Figure 7, which were obtained using Equations (1) and (2). Thus, the strength development of mortar can be predicted efficiently at temperatures lower than 0 °C, even when the antifreezing admixture is added.

Based on the above results, when the antifreezing admixture is added, the maturity at temperatures lower than 0 °C considering the decrease in the freezing temperature can be expressed as shown in Figure 13. Moreover, the maturity considering the addition of the antifreezing admixture can expressed as Equations (12) and (13).

## 4. Conclusions

This study investigated the effect of temperature and time at temperatures lower than 0 °C on mortar mixed with antifreezing admixture to determine the maturity equation with the aim of expressing the effect universally, and the following conclusions were obtained.

(1) This study verified that the maturity equation proposed in previous studies for temperatures lower than 0 °C can be used for BB as well.

(2) The effect of the addition of the antifreezing admixture on the strength development of mortar can be expressed in terms of the reference temperature and dependence of the chemical potential reduction of water on the reaction rate.

(3) A new maturity equation considering the addition of the antifreezing admixture was proposed for temperatures lower than 0 °C as follows:Case of *T* < −0.28*x*
*M_f_* = ∑(10 − 0.28*x*) × exp{(−0.60 + 0.038*x*) × (−*T_c_* − 0.28*x*)^(0.74 − 0.038*x*)^}·*Δt*
where *M_f_* denotes the maturity (°C) at temperature *T_c_*, *Δt* denotes the time interval (days), and *x* denotes the quantity (ℓ) of the antifreezing admixture added per 100 kg of cement.

(4) The new maturity equation, considering the influence of the antifreezing admixture in strength development, can be expressed, and it can be calculated of the strength development for each addition rate of the antifreezing admixture in case of temperatures lower than 0 °C.

## Figures and Tables

**Figure 1 materials-12-03172-f001:**
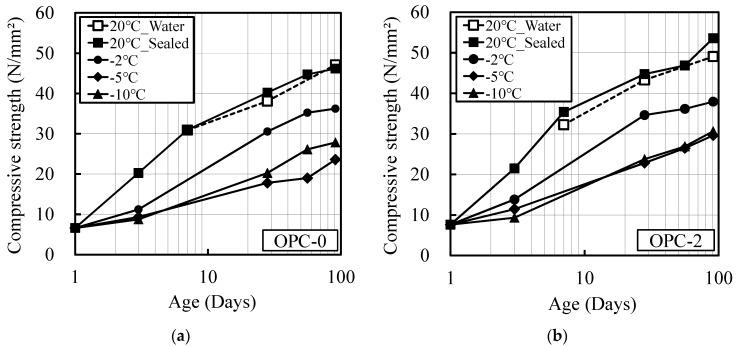
Relationship between compressive strength and age (OPC), (**a**) OPC-0; (**b**) OPC-2; (**c**) OPC-4; (**d**) OPC-6.

**Figure 2 materials-12-03172-f002:**
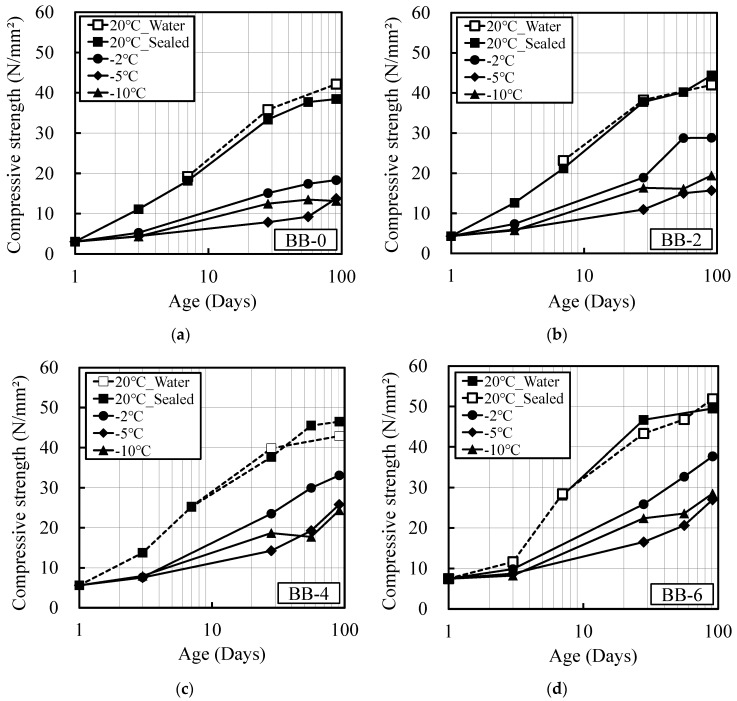
Relationship between compressive strength and age (BB), (**a**) BB-0; (**b**) BB-2; (**c**) BB-4; (**d**) BB-6.

**Figure 3 materials-12-03172-f003:**
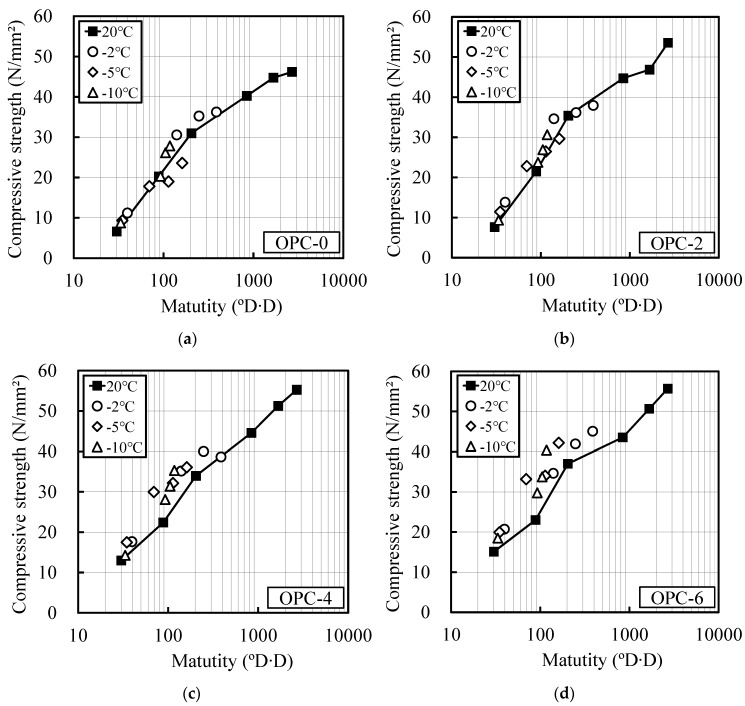
Relation between maturity and compressive strength (OPC): Previous equation for maturity, (**a**) OPC-0; (**b**) OPC-2; (**c**) OPC-4; (**d**) OPC-6.

**Figure 4 materials-12-03172-f004:**
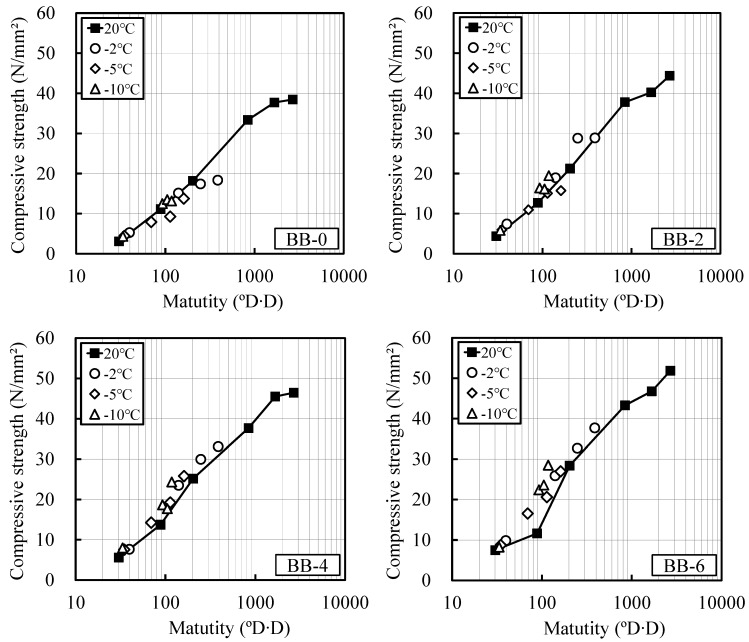
Relation between maturity and compressive strength (BB): Previous equation for maturity, (**a**) BB-0; (**b**) BB-2; (**c**) BB-4; (**d**) BB-6.

**Figure 5 materials-12-03172-f005:**
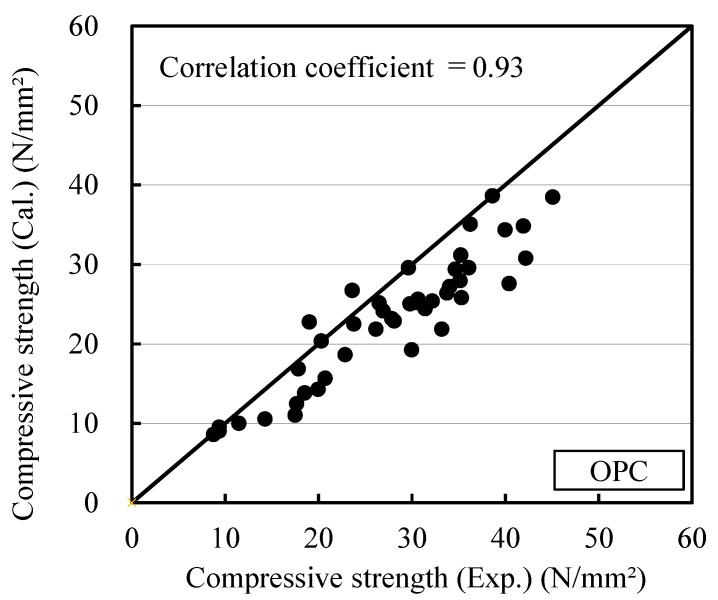
Measured compressive strength and calculated compressive strength (OPC): Previous equation for maturity.

**Figure 6 materials-12-03172-f006:**
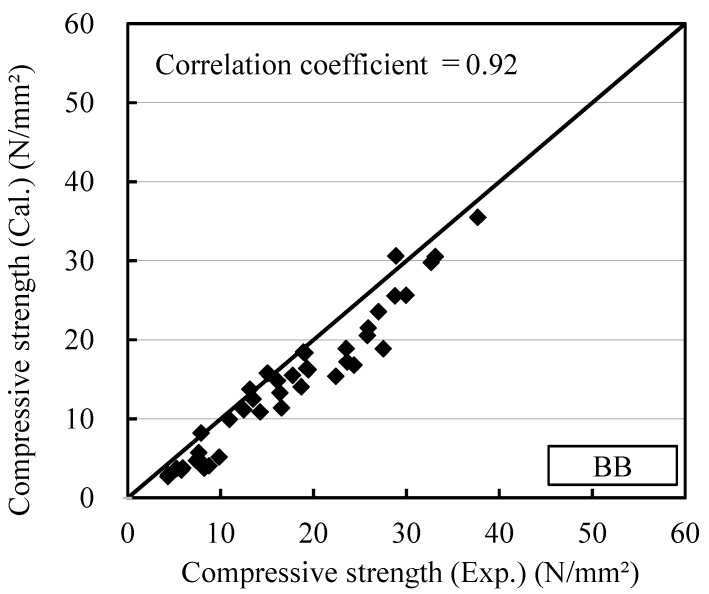
Measured compressive strength and calculated compressive strength (BB): Previous equation for maturity.

**Figure 7 materials-12-03172-f007:**
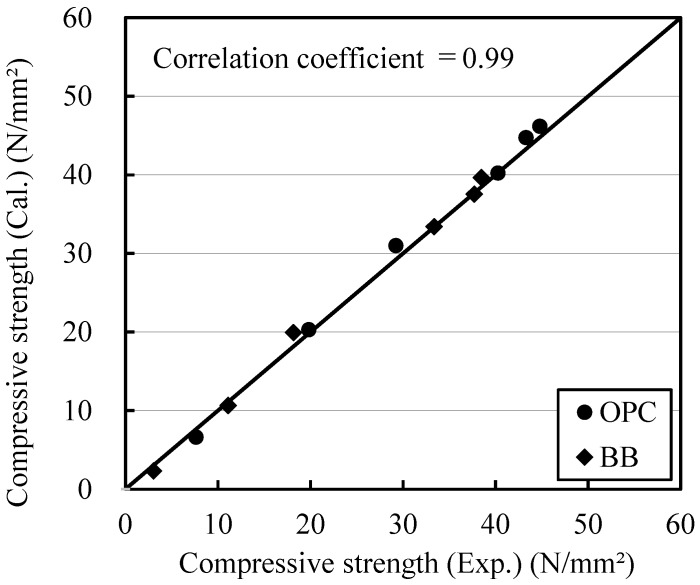
Relationship between the measured compressive strength and calculated compressive strength.

**Figure 8 materials-12-03172-f008:**
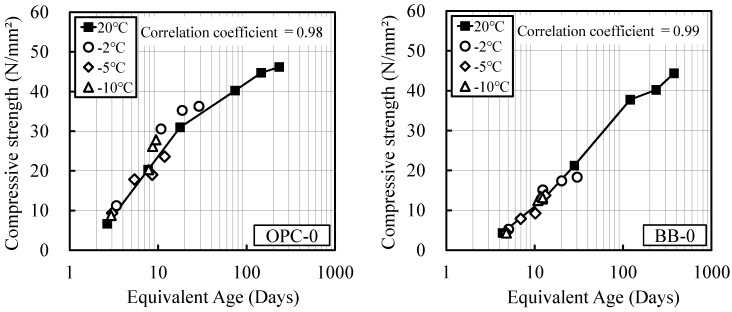
Relationship between equivalent age and measured compressive strength considering freezing.

**Figure 9 materials-12-03172-f009:**
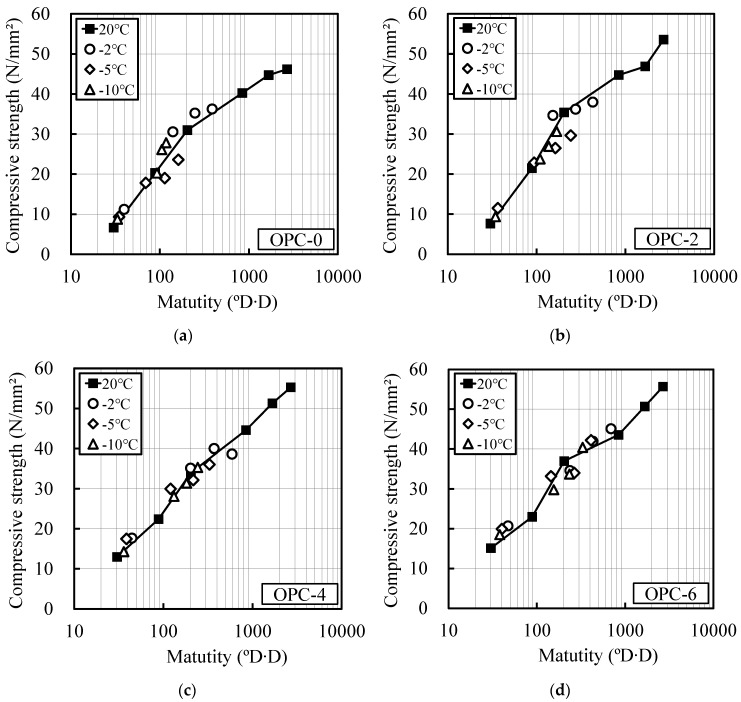
Relation between maturity and compressive strength (OPC): Proposal equation for this study, (**a**) OPC-0; (**b**) OPC-2; (**c**) OPC-4; (**d**) OPC-6.

**Figure 10 materials-12-03172-f010:**
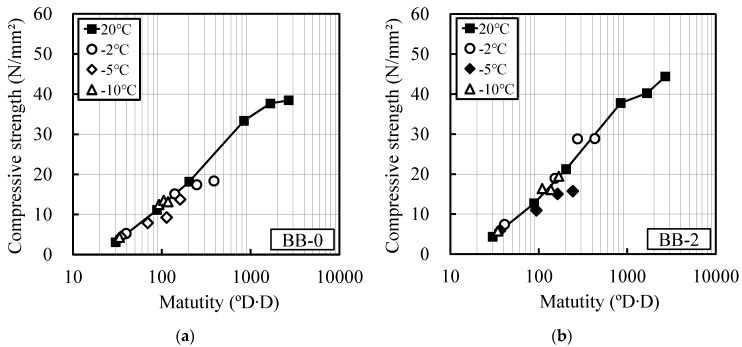
Relation between maturity and compressive strength (BB): Proposal equation for this study, (**a**) BB-0; (**b**) BB-2; (**c**) BB-4; (**d**) BB-6.

**Figure 11 materials-12-03172-f011:**
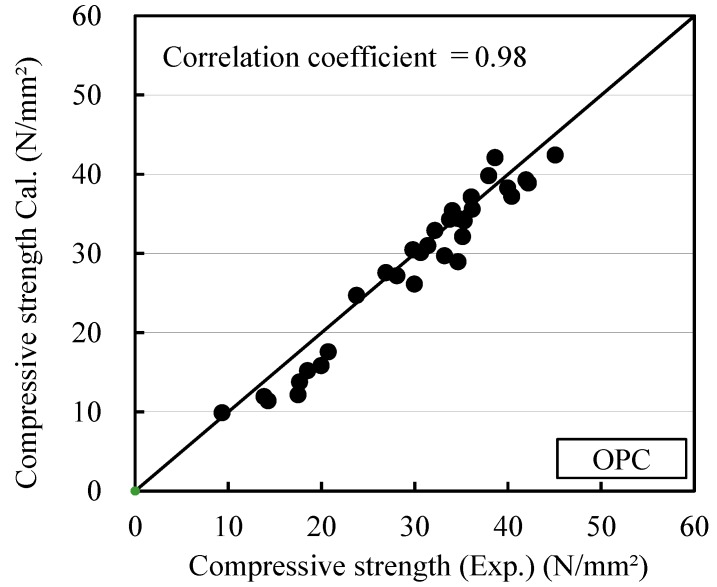
Measured compressive strength and calculated compressive strength (OPC): Proposal equation for this study.

**Figure 12 materials-12-03172-f012:**
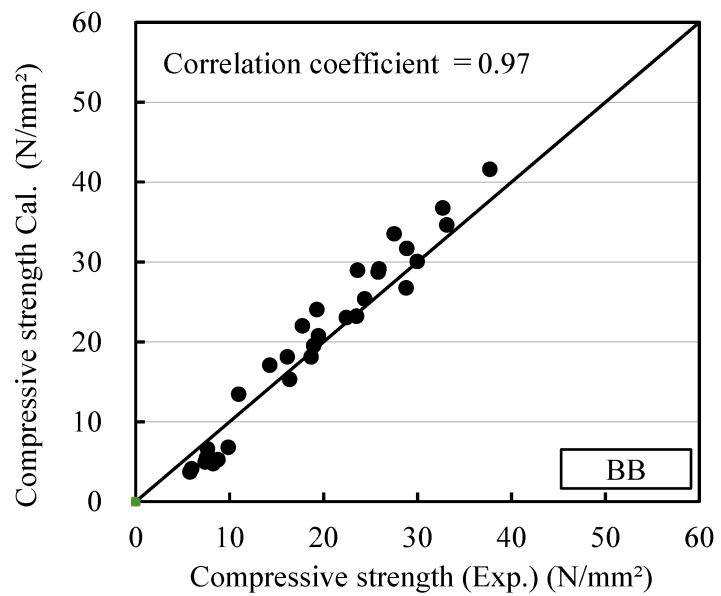
Measured compressive strength and calculated compressive strength (BB): Proposal equation for this study.

**Figure 13 materials-12-03172-f013:**
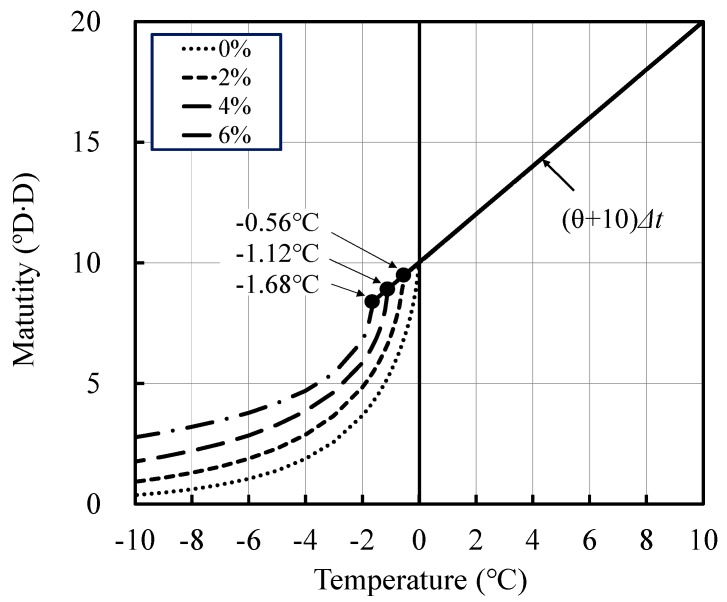
Relationship between the measured compressive strength and calculated compressive strength.

**Table 1 materials-12-03172-t001:** Experiment plan.

Cement	Antifreezing Admixture	W/C (%)	C:S	Placement Temperature (°C)	Curing Temperature (°C)	Curing Condition	Test Item
	Rate of Addition (ℓ/C = 100 kg)						
OPC	0, 2, 4, 6	50	1:3	20	20 * 20→−2 20→−5 20→−10	Sealed curing Water curing *	Compressive strength Temperature
BB							

**Table 2 materials-12-03172-t002:** Materials Used.

Type	
Cement	Ordinary Portland cement, Density: 3.15 g/cm^3^, Specific surface area: 3490 cm^3^/g
	Blast furnace slag cement, Density: 2.91 g/cm^3^, Specific surface area: 3250 cm^3^/g
Sand	Pit sand (siliceous sand), Density: 2.69 g/cm^3^, Absorption ratio: 1.52%, Unit volumetric mass: 1.86 kg/l, Actual ratio: 70.2%, Fineness modulus: 2.70
Admixture	Antifreezing admixture (Main component: inorganic nitrogenous compound; Nitrite, Nitrate), Density: 1.41–1.45 g/cm^3^, Amount of chloride ion: 0.01%, Total alkali amount: 0%

**Table 3 materials-12-03172-t003:** Binding materials properties.

Binder	Specific Surface Area (g/cm^3^)	Density (g/cm^3^)	Chemical Composition (%)							
			SiO_2_	Al_2_O_3_	Fe_2_O_3_	CaO	MgO	SO_3_	f-CaO	Ig.loss
OPC	3490	3.15	21.4	5.5	2.9	64.3	1.9	1.8	0.25	0.68
BB	3250	3.05	26.1	8.9	2.0	55.6	3.8	-	-	0.68

**Table 4 materials-12-03172-t004:** Sieve analysis test of sand.

**Type**	**Cumulative Passing (%)**							**Fineness Modulus** **(F.M.)**
Sieve Size (mm)	0.15	0.3	10	1.2	2.5	5	10	
Sand	2.7	23.1	100	67.3	87.0	99.4	100	2.70

**Table 5 materials-12-03172-t005:** Experimental coefficients (a,b).

Antifreezing Admixture (ℓ/C = 100 kg)	Equivalent Age				Maturity			
	OPC		BB		OPC		BB	
	a	b	a	b	a	b	a	b
0	−1.81	−0.55	−2.87	−0.62	−18.0	−0.66	−30.4	−0.68
2	−1.88	−0.66	−2.74	−0.64	−17.6	−0.66	−28.1	−0.67
4	−1.69	−0.54	−2.48	−0.53	−14.0	−0.60	−20.8	−0.61
6	−1.46	−0.54	−2.84	−0.58	−9.5	−0.55	−33.8	−0.72

**Table 6 materials-12-03172-t006:** Experimental coefficients (α).

Type	OPC	BB
*α*	31.2	30.4

**Table 7 materials-12-03172-t007:** Experimental coefficients (c, d).

Antifreezing Admixture (ℓ/C = 100 kg)	*c*	*d*	
	OPC and BB	OPC	BB
2	13.2	1.01	1.32
4	6.2	1.09	1.39
6	3.6	1.01	1.30

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
