# Peer review of "Maturity and Strength Development of Mortar with Antifreezing Admixture at Temperatures Lower than 0 °C"

_materials, 2019, doi:10.3390/ma12193172_

Round 1
Reviewer 1 Report
Blast furnace slag and antifreezing admixtures are generally technically mutually exclusive. Slag is used to reduce the heat of hydration and increase chemical resistance, however, due to the slow increase in strength, we limit its use in the winter.
The tests carried out on mortars raise doubts, especially when exposed to negative temperatures, due to the differences in the heat of aggregate and paste hydration.
The w/c ratio is given in %. It is customary to use the decimal fraction.
The antifreeze admixture is given in %, according to the standard it is given as % of cement mass
The term 'curing' is used to determine the curing of concrete. Applied 'maturity' is used in the maturation of people.
Applied BB cement with the addition of blast furnace slag during empirical analyzes raises doubts due to the lack of information regarding the slag content in cement.
Reviewer 2 Report
COMMENTS TO AUTHORS
The research developed in this work is very interesting. The methodology used in the study of the process of setting and hardening of mixtures of cement mortar, dosed with antifreeze additives, is adequate.
The mathematical model developed follows a logical-deductive process appropriate to this type of research and is based on prior consolidated research, based on the Arrhenius model for the dependence of a temperature reaction. However, I propose to the Authors some suggestions with the aim of improving the manuscript for possible publication in Materials. Are the following:
2.1 Experimental design and methods
The authors indicate that for the experimental phase they use a mortar with water – cement ratio of 50%. (Line 53)
Although it is perfectly understood, in the mortars and concretes dosing processes, the relationship between two components, in this case water and cement is a number, not a percentage. According to that criterion, I should say "... water-cement ratio of 0.5". On the other hand, it is indicated that the “…cement–fine aggregate ratio of 1:3…” (It is not said, but the relationship will be by weight. It should indicate)
The dosage by weight is 1:3:0.5 (cement/fine aggregate/water). This dosage is the standard for the determination of the mechanical resistance characteristic of Portland cements. Mention should be made of the standard
Regarding the fine aggregate dosed for the test samples, Table 2 shows some characteristics, but no reference is made to its nature (siliceous sand, limestone sand etc.), sieve sizes, particle size module. Table 2 should be improved by providing more characteristics of the materials used.
The Authors indicate that they add different doses of antifreeze additive (0, 2, 4, and 6) % per 100 kg of cement added. Would it not be more correct to indicate that the antifreeze is dosed at (0, 2, 4 and 6) % with respect to the weight of cement dosed?
On Line 59, where it says "... cylindrical specimens measuring φ50 × 100 mm were fabricated ...” replace (φ50) for (Ø50).
4. Conclusions
The developed research allows extending the final conclusions. I believe that the Authors should be more explicit in their conclusions, since they have sufficient information for these.
References (Bibliography)
Authors should adapt the bibliography to the Materials format. There are many wrongs about it. Please, review it and read the Authors Guide
Tables
The Authors must to make a new numbering of the Tables, since there are two "Table 2" (Line 57 and Line 132)
Important, they should also adapt the Tables to the Material model, eliminating vertical lines.
Author Contributions:
The Authors must indicate the responsibilities assumed in the investigation developed in the document.
Final comment
I congratulate the Authors for the work done and encourage them to continue their researches
Round 2
Reviewer 1 Report
I accept the corrections and responses of the authors.